# Hyperbolic Contrastive Learning for Visual Representations beyond Objects

## Abstract

Despite the rapid progress in visual representation learning driven by self-/un-supervised methods, both objects and scenes have been primarily treated using the same lens. In this paper, we focus on learning representations for objects and scenes explicitly in the same space. Motivated by the observation that visually similar objects are close in the representation space, we argue that the scenes and objects should further follow a hierarchical structure based on their compositionality. To exploit such a structure, we propose a contrastive learning framework where a Euclidean loss is used to learn object representations and a hyperbolic loss is used to regularize scene representations according to the hierarchy. This novel hyperbolic objective encourages the scene-object hypernymy among the representations by optimizing the magnitude of their norms. We show that when pretraining on the COCO and OpenImages datasets, the hyperbolic loss improves downstream performance across multiple datasets and tasks, including image classification, object detection, and semantic segmentation. We also show that the properties of the learned representations allow us to solve various vision tasks that involve the interaction between scenes and objects in a zero-shot way.

## 1 Introduction

Our visual world is diverse and structured. Imagine taking a close-up of a box of cereal in the morning. If we zoom out slightly, we may see different nearby objects such as a bowl of milk, a cup of hot coffee, today's newspaper, or reading glasses. Zooming out further, we will probably recognize that these items are placed on a dining table with the kitchen as background rather than inside a bathroom. Such scene-object structure is diverse, yet not completely random. In this paper, we aim at learning visual representations of both the cereal box (objects) and the entire dining table (scenes) in the same space while preserving such hierarchical structures.

Un-/self-supervised learning has become a standard method to learn visual representations [27, 12, 25, 13, 6, 7, 49]. Although these methods attain superior performance over the supervised pretraining on object-centric datasets such as ImageNet [25, 6], inferior results are observed on images depicting multiple objects such as OpenImages or COCO [67]. Several methods have been proposed to mitigate this issue [67, 68, 37, 1], but all focus on learning improved object representations or dense pixel representations, instead of explicitly modeling the representations for scene images. The object representations learned by these methods present a natural topology [66]. That is, the objects from visually similar classes lie close to each other in the representation space. However, it is not clear how the representations of scene images should fit into that topology. Naively applying existing contrastive learning results in sub-optimal topology of scenes and objects as well as unsatisfactory performance as we will show in the experiment. To this end, we argue that a hierarchical structure can be naturally adopted. Considering scenes as the composition of different kinds of objects, we

can construct a forest structure to describe such relationships, where the root nodes are the visually similar objects, and the scene images consisting of them are placed as the descendants. We call this structure the object-centric scene hierarchy.

The intermediate modeling difficulty induced by this structure is the combinatorial explosion. A finite number of objects can lead to exponentially many kinds of scenes due to the composition. Hyperbolic space is known for its provably better capacity in modeling infinite trees compared with Euclidean space [21, 26, 34]. Therefore, we propose to employ a hyperbolic objective to regularize the scene representations. Our framework builds upon MoCo [27], which has been shown to learn good object representations. To learn representations of scenes, we sample the co-occurring scene-object pairs as the positive pairs, and objects that are not part of that scene as the negative samples, and use these pairs to compute an auxiliary hyperbolic contrastive objective. Our model is trained to reduce the distance between positive pairs and push away the negative pairs in a hyperbolic space.

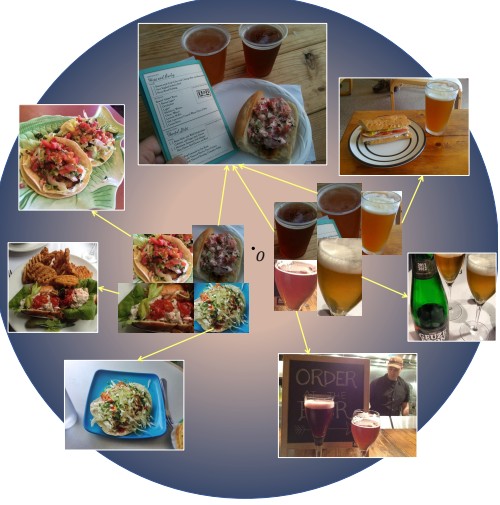

Figure 1: Illustration of the representation space learned by our models. Object images of the same class tend to gather near the center around similar directions, while the scene images are far away in these directions with larger norms.

Contrastive learning models generally compute their objectives on a hypersphere [27, 12]. By discarding the norm information, these models effectively circumvent the shortcut of minimizing objectives by tuning the norms and obtain better downstream performance. At the same time, they also lose control of the representative power in the magnitude of the norm and leave the images disorganized. However, in hyperbolic space, it is the magnitude of the norm that is used to model the hypernymy of the hierarchical structure [43, 58, 51]. When projecting the representations to the hyperbolic space, the norm information is preserved and used to determine the Riemannian distance, which eventually affects our loss. Since the hyperbolic space is diffeomorphic and conformal to the Euclidean, our hyperbolic contrastive loss is completely differentiable and complementary to the original contrastive objective.

When training simultaneously with the original contrastive objective for objects and our proposed hyperbolic contrastive objective for scenes, the resulting representation space exhibits the desired hierarchical structure while keeping the object clustering topology intact as shown in Figure 1. We demonstrate the effectiveness of the learned representations on several downstream tasks, from image classification to object detection. We also show that the properties possessed by the representations allow us to perform various vision tasks in a zero-shot way, from label uncertainty quantification to out-of-context object detection. Our contributions are summarized below:

1. We propose to learn representations for both object and scene images simultaneously using un-/self-supervised methods. We identify an object-centeric scene hierarchy that the representations are expected to follow.

2. We propose a framework with a novel hyperbolic contrastive loss to regularize the scene representations with positive and negative pairs sampled from the hierarchy.

3. We show that the magnitude of representation norms effectively reflect the scene-objective hypernymy, and such representations transfer better to multiple downstream tasks.

## 2   Method

In this section, we elaborate our approach to learn visual representations of object and scene images. We start with describing the hierarchical structure between objects and scenes.

## 2.1 Object-Centric Scene Hierarchy

From simple object co-occurrence statistics [20, 39] to finer object relationships [29, 31], using hierarchical relationships between objects and scenes to understand images is not new. Previous studies primarily work on an instance-level hierarchy by dividing an image into its lower-level elements recursively - a scene contains multiple objects, an object has different parts, and each part may consist of even lower-level features [47, 46, 15]. While this is intuitive, it describes a hierarchical structure contained in the individual images. In our task, we would like to work on the structure from the view of the entire dataset to learn a representation space shared by objects and scenes. To this end, we argue that it is more natural to consider an object-centric hierarchy.

It is known that when training an image classifier, though not being optimized directly, the objects from visually similar classes often lie close to each other in the representation space [66], which has become the cornerstone of contrastive learning [27, 12]. Motivated by this observation, we believe that the representation of each scene image should also be close to the object clusters it consists of. However, they require a much larger volume due to the exponential number of possible compositions. Another way to think about the object-centric hierarchy is through the generality and specificity as often discussed in the language literature [40, 43]. An object concept is general when standing alone in the visual world, and it will become specific when a certain context is given. For example, "a desk" is thought to be a more general concept than "a desk in a classroom with a boy sitting on it".

Therefore, we propose to study an object-centric hierarchy across the entire dataset. Formally, given a set of images $\mathcal{S} = \{s_1, s_2, \cdots, s_n\}$, $\mathcal{O}_i = \{o_i^1, o_i^2, \cdots, o_i^{n_i}\}$ are the object bounding boxes contained in the image $s_i$. We define the regions of scene $\mathcal{R}_i = \{r_i^1, r_i^2, \cdots, r_i^{m_i}\}$ to be partial areas of the image $s_i$ that contain multiple objects such that $r_i^j = \cup_k o_i^k$, where $o_i^k \in \mathcal{O}_i$ and object $k$ is in the region $j$. We define the object-centric forest $T = (V, E)$ to be that $V = \mathcal{S} \cup \mathcal{O} \cup \mathcal{R}$, where $\mathcal{R} = \mathcal{R}_1 \cup \cdots \cup \mathcal{R}_n$ and $\mathcal{O} = \mathcal{O}_1 \cup \cdots \cup \mathcal{O}_n$. For $u, v \in V$, $e = (u, v)$ is an edge of $T$ if $u \subseteq v$ or $v \subseteq u$. Note that the natural scene images $\mathcal{S}$ are always put as the leaf nodes.

## 2.2 Representation Learning beyond Objects

To describe our proposed model that is built on this hierarchy, we begin with a brief review of the hyperbolic space and its several properties that will be used in our model. For comprehensive introductions to the Riemannian geometry and hyperbolic space, we refer the readers to [32, 17].

### 2.2.1 Hyperbolic Space

A hyperbolic space $(\mathbb{H}^m, g)$ is a complete, connected Riemannian manifold with constant negative sectional curvature. These special manifolds are all isometric to each other with the isometries defined as $O^+(m, 1)$. Among these isometries, there are five common models that previous studies often work on [5]. In this paper, we choose the Poincaré ball $\mathbb{D}^n := \{p \in \mathbb{R}^n \mid \|p\|^2 < r^2\}$ as our basic model [43, 58, 22], where $r > 0$ is the radius of the ball. The Poincaré ball is coupled with a Riemannian metric $g_{\mathbb{D}}(p) = \frac{4}{(1 - \|p\|^2/r^2)^2} g_{\mathbb{E}}$, where $p \in \mathbb{D}^n$ and $g_{\mathbb{E}}$ is the canonical metric of the Euclidean space. For $p, q \in \mathbb{D}$, the Riemannian distance on the Poincaré ball induced by its metric $g_{\mathbb{D}}$ is defined as follows:

$$d_{\mathbb{D}}(p, q) = 2r \tanh^{-1} \left( \frac{\|-p \oplus q\|}{r} \right), \tag{1}$$

where $\oplus$ is the Möbius addition and it is clearly differentiable. In addition, the Poincaré ball can be viewed as a natural counterpart of the hypersphere as it allows all directions, unlike the other models such as the halfspace or hemisphere models that have constraints on the directions. The hyperbolic space is globally differomorphic to the Euclidean space, which is stated in the theorem below:

**Theorem 1.** *(Cartan–Hadamard). For every point $p \in \mathbb{H}^n$ the exponential map $\exp_p : T_p\mathbb{H}^n \approx \mathbb{R}^n \to \mathbb{H}^n$ is a smooth covering map. Since $\mathbb{H}^n$ is simply connected, it is diffeomorphic to $\mathbb{R}^n$.*

Specifically, for $p \in \mathbb{D}^n$ and $v \in T_p\mathbb{D}^n \approx \mathbb{R}^n$, the exponential map of the Poincaré ball $\exp_p : T_p\mathbb{D}^n \to \mathbb{D}^n$ is defined as

$$\exp_p(v) := p \oplus \left( \tanh \left( \frac{r\|v\|}{r^2 - \|p\|^2} \right) \frac{rv}{\|v\|} \right), \tag{2}$$

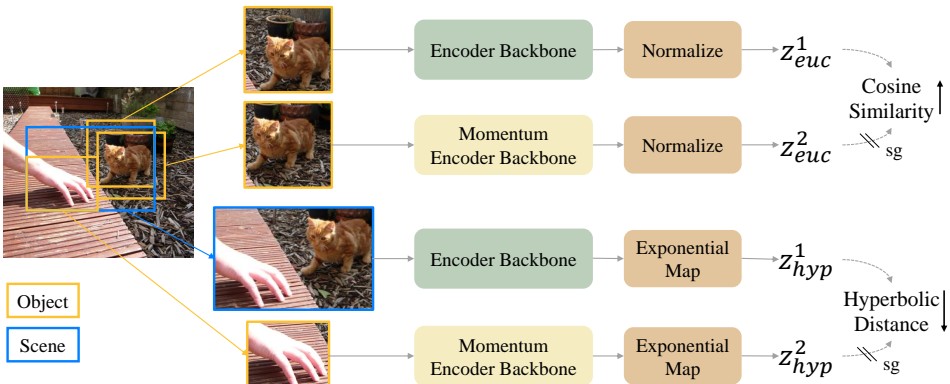

Figure 2: Our **H**yperbolic **C**ontrastive **L**earning (HCL) framework has two branches: given a scene image, two object regions are cropped to learn the object representations with a loss defined in the Euclidean space focusing on the representation directions. A scene region as well as a contained object region are used to learn the scene representations with a loss defined in the hyperbolic space that affects the representation norms.

The exponential map gives us a way to map the output of a network, which is in the Euclidean space, to the Poincaré ball. In practice, to avoid numerical issues, we clip the maximal norm of $v$ with $r - \varepsilon$ before the projection, where $\varepsilon > 0$. During the backpropagation, we perform RSGD [4] by scaling the gradients with $g_{\mathbb{D}}(p)^{-1}$. Intuitively, this forces the optimizer to take a smaller step when $p$ is closer to the boundary. The scaling factor is lower bounded by $\mathcal{O}(\varepsilon^2)$.

The immediate consequence of the negative curvature is that for any point $\boldsymbol{p} \in \mathbb{H}^m$, there are no conjugate points along any geodesic starting from $\boldsymbol{p}$. Therefore, the volume grows exponentially faster in hyperbolic space than in Euclidean space. Such a property makes it suitable to embed the hierarchical structure that has constant branching factors and exponential number of nodes. This is formally stated in the theorem below:

**Theorem 2.** *[21, 26] Given a Poincaré ball $\mathbb{D}^n$ with an arbitrary dimension $n \geq 2$ and any set of points $p_1, \cdots, p_m \in \mathbb{D}^n$, there exists a finite weighted tree $(T, d_T)$ and an embedding $f : T \to \mathbb{D}^n$ such that for all $i, j$,*

$$\left| d_T \left( f^{-1}\left(x_i\right), f^{-1}\left(x_j\right) \right) - d_{\mathbb{D}}\left(x_i, x_j\right) \right| = \mathcal{O}(\log(1 + \sqrt{2}) \log(m))$$

Intuitively, the theorem states that any tree can be embedded into a Poincaré disk ($n = 2$) with low distortion. On the contrary, it is known that the Euclidean space with unbounded number of dimensions is not able to achieve such a low distortion [34]. One useful intuition [51] to help understand the advantage of the hyperbolic space is given two points $p, q \in \mathbb{D}^n$ s.t. $\|p\| = \|q\|$,

$$d_{\mathbb{D}}(p, q) \to d_{\mathbb{D}}(p, 0) + d_{\mathbb{D}}(0, q), \text{ as } \|p\| = \|q\| \to r \tag{3}$$

This property basically reflects the fact that the shortest path in a tree is the path through the earliest common ancestor, and it is reproduced in the Poincaré when points are both close to the boundary.

### 2.2.2 Hyperbolic Contrastive Learning

With the theoretical benefits of the hyperbolic space stated above, we propose a contrastive learning framework as shown in Figure 2. We adopt two losses to learn the object and scene representations. First, as shown in the top branch of Figure 2, we crop two views of a jittered and slightly expanded object region as the positive pairs and feed into the base and momentum encoders to calculate the object representations. We denote the output after the normalization to be $\mathbf{z}_{\text{euc}}^1$ and $\mathbf{z}_{\text{euc}}^2$. Considering the computational cost of large batch sizes, we follow MoCo [27, 14] to leverage a memory bank to store the negative representations $\boldsymbol{z}_{\text{euc}}^n$ which are the features $\mathbf{z}_{\text{euc}}^2$ from the previous batches. The Euclidean loss for this image is then calculated as:

$$\mathcal{L}_{\text{euc}} = -\log \frac{\exp\left(\mathbf{z}_{\text{euc}}^1 \cdot \mathbf{z}_{\text{euc}}^2 / \tau\right)}{\exp\left(\mathbf{z}_{\text{euc}}^1 \cdot \mathbf{z}_{\text{euc}}^2 / \tau\right) + \sum_n \exp\left(\mathbf{z}_{\text{euc}}^1 \cdot \mathbf{z}_{\text{euc}}^n / \tau\right)},$$

where $\tau$ is a temperature parameter.

While this loss aims at learning object representations, we also design a hyperbolic contrastive objective to learn the representations for scene images. We sample the positive region pairs $u$ and $v$ from object-centric scene hierarchy $T$ such that $(u, v) \in E$. In other words, as shown in the bottom branch of Figure 2, the objects contained in one region are required to be a subset of the objects in the other. We sample the negative samples of $u$ to be $\mathcal{N}_u = \{v | (u, v) \notin E\}$. However, building and sampling from the entire hierarchy explicitly is slow and memory consuming. Instead, according to the assumption that there are exponentially more scenes than object classes in practice, given a scene image $s$, we always sample $u \in \mathcal{R} \cup \{s\}$ to be a scene region, $v \in \mathcal{O}$ to be an object that occurs in $u$, and $\mathcal{N}_u$ to be the other objects that are not in $u$.

The pair of scene and object images are fed into the base and momentum encoders that share the weights with the Euclidean branch. However, instead of normalizing the output of the encoders, we use the exponential map defined in the equation (2) to project these features in the Euclidean space to the Poincaré ball, which are denoted as $\mathbf{z}_{\text{hyp}}^1$ and $\mathbf{z}_{\text{hyp}}^2$. Further, we replace the inner product in the cross-entropy loss with the negative hyperbolic distance as defined in equation (1). We calculate the hyperbolic contrastive loss as follows:

$$\mathcal{L}_{\text{hyp}} = -\log \frac{\exp\left(-d_{\mathbb{D}}(\mathbf{z}_{\text{hyp}}^1, \mathbf{z}_{\text{hyp}}^2)/\tau\right)}{\exp\left(-d_{\mathbb{D}}(\mathbf{z}_{\text{hyp}}^1, \mathbf{z}_{\text{hyp}}^2)/\tau\right) + \sum_n \exp\left(-d_{\mathbb{D}}(\mathbf{z}_{\text{hyp}}^1, \mathbf{z}_{\text{hyp}}^n)/\tau\right)},$$

When minimizing the distances of all the positive pairs, With the intuition from Equation (3), it would be beneficial to put the nodes near the root close to the center to achieve a overall lower loss. The overall loss function of our model is as follows:

$$\mathcal{L} = \mathcal{L}_{\text{euc}} + \lambda \mathcal{L}_{\text{hyp}},$$

where $\lambda$ is an scaling parameter to control the trade-off between hyperbolic and Euclidean losses.

## 3 Experiments

### 3.1 Implementation Details

**Pre-training phase.** We pre-train our method on two datasets: COCO [33] and a subset of Open-Images [41]. Both of these datasets are multi-object datasets; OpenImages [41] ($\sim$ 212k images) contains 12 objects on average per image and COCO ($\sim$ 118k) contains 6 objects on average. We experiment with both the ground truth bounding box (GT) and using selective search [60] following the previous method [67] (SS) to acquire objects. For the optimizer setups and augmentation recipes, we follow the standard protocol described in MoCo-v2 [14] unless denoted otherwise. We find that a base learning rate of 0.3 works better for us as compared to 0.03. We adopt the linear learning rate scaling receipt that $lr = 0.3 \times \text{BatchSize}/256$ [24] and batch size of 128 by default on 4 NVIDIA p6000 gpus. To ensure fair comparison, we also pre-train the baselines with a learning rate of 0.3. We train our models on both datasets for 200 epochs. For the hyperparameters of our hyperbolic objective, we use $r = 4.5$, $\lambda = 0.1$, and $\varepsilon = 1e^{-5}$. More details on the OpenImages dataset as well as training setups can be found in Appendix A.

**Downstream tasks.** We evaluate our pre-trained models on image classification, object-detection and semantic segmentation. For classification, we show linear evaluation (lineval) accuracy, i.e we freeze the backbone and only train the final fc layer. We test on VOC [19], ImageNet-100 [57] and ImageNet-1k [16] datasets. To test the discriminative capacity of the representations on both objects and scenes, we create a dataset by mixing the ImageNet-100 and a subset of Place-205 [70] datasets, which we refer to as the INPMix dataset. More details of this dataset can be found in Appendix A. For object detection and semantic segmentation, we show results on the COCO and Pascal VOC `trainval2017` datasets. For VOC object detection, COCO object detection and COCO semantic segmentation, we closely follow the common protocols listed in Detectron2 [65].

| | Pre-train dataset | Bbox type | VOC | IN-100 | INPMix | IN-1k |
|---|---|---|---|---|---|---|
| MoCo-v2 | COCO | - | 64.79 | 64.84 | 41.83 | 51.17 |
| HCL/$\mathcal{L}_{\text{hyp}}$ | COCO | SS | 73.13 | 73.84 | 51.28 | 54.21 |
| HCL/$\mathcal{L}_{\text{hyp}}$ | COCO | GT | 75.55 | 76.22 | 51.25 | 54.52 |
| HCL | COCO | SS | 74.19 | 75.16 | 51.35 | 55.03 |
| HCL | COCO | GT | **76.51** | **76.74** | **51.63** | **55.63** |
| MoCo-v2 | OpenImages | - | 69.95 | 72.80 | 49.59 | 54.12 |
| HCL/$\mathcal{L}_{\text{hyp}}$ | OpenImages | GT | 73.79 | 77.36 | 52.96 | 57.57 |
| HCL | OpenImages | SS | 74.31 | 78.14 | 53.21 | 58.12 |
| HCL | OpenImages | GT | **75.40** | **79.08** | **53.82** | **58.51** |

Table 1: **Classification results with linear evaluation.** Our model improves scene-level classification on the VOC [19] and INPMix [70] datasets, and object-level classification on ImageNet-100 [57] and ImageNet-1k [16] datasets.

| Detection | Dataset | AP | AP$_{50}$ | AP$_{75}$ |
|---|---|---|---|---|
| MoCo-v2 | COCO | 34.6 | 53.5 | 37.0 |
| HCL/$\mathcal{L}_{\text{hyp}}$ | COCO | 36.1 | 55.2 | 37.9 |
| HCL | COCO | **37.0** | **56.1** | **39.8** |
| MoCo-v2 | VOC | 51.5 | 79.4 | 56.1 |
| HCL - $\mathcal{L}_{\text{hyp}}$ | VOC | 53.7 | 80.5 | 59.4 |
| HCL | VOC | **54.4** | **81.4** | **60.2** |
| Segmentation | Dataset | AP$_{\text{s}}$ | AP$_{\text{l}}$ | AP$_{\text{m}}$ |
| MoCo-v2 | COCO | 30.4 | 50.1 | 32.3 |
| HCL/$\mathcal{L}_{\text{hyp}}$ | COCO | 31.5 | 52.0 | 33.8 |
| HCL | COCO | **32.5** | **52.9** | **34.6** |

Table 2: **Object detection and Semantic Segmentation results.** Our model improves on both tasks on COCO [33] and VOC [19] datasets.

## 3.2  Main Results

This section discusses our main results on the downstream image classification, object detection, and semantic segmentation tasks. As the goal of this paper is not to present another state-of-the-art self-supervised learning method, we primarily compare with the backbone model MoCo-v2 [27]. Another important baseline we consider is our model without the hyperbolic loss $\mathcal{L}_{\text{hyp}}$; therefore only the object representations are learned, which we denote as HCL/$\mathcal{L}_{\text{hyp}}$.

**Image classification.** As shown in Table 1, HCL improves image classification on both scene-level datasets (VOC and INPMix) and object-level datasets (ImageNet). When pretraining on OpenImages, HCL improves ImageNet lineval accuracy by 0.94% and VOC lineval classification accuracy by 1.61 mAP. We observe similar improvements when pretraining on COCO. HCL improves accuracy whether we use ground truth object bounding boxes or boxes generated by selective search. In general, we observe a larger improvement of using HCL on OpenImages than COCO, which supports our observation that HCL would improve more on the dataset with more objects per images.

**Object detection and semantic segmentation.** Table 2 reports the object detection and semantic segmentation results using Mask R-CNN, following [14]. It shows consistent improvements over the baselines on VOC object detection, COCO object detection, and COCO semantic segmentation.

## 3.3  Properties of Models Trained with HCL

The visual representations learned by HCL have several useful properties. In this section, we evaluate the representation norm as an measure of the label uncertainty for image classification datasets, and evaluate the object-scene similarity in terms of out-of-context detection.

### 3.3.1  Label Uncertainty Quantification

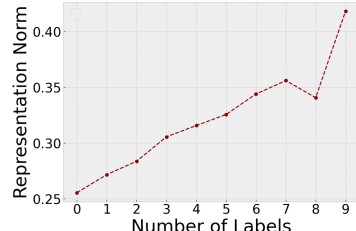

Figure 4: Average representation norms of images with different number of labels in ImageNet-ReaL [3].

| Method | Indicator | Datasets | |
|---|---|---|---|
| | | IN-Real | COCO |
| MoCo | Entropy | 0.633 | 0.791 |
| Supervised | Entropy | 0.671 | 0.793 |
| HCL | Norm | 0.655 | **0.839** |
| Ensemble | Entropy+Norm | **0.717** | 0.823 |

Table 3: NDCG scores of the image rankings based on the different indicators and models, and evaluated by the the number of labels per image.

ImageNet [16] is an image classification dataset consisting of object-centered images, each of which has a single label. As the performance on this dataset gradually saturated, the original labels have been scrutinized more carefully [50, 59, 54, 3, 61]. Prevailing labeling issues in the validation set

Smallest norms (objects) ◄────── ■■■ ──────► Largest norms (scenes)

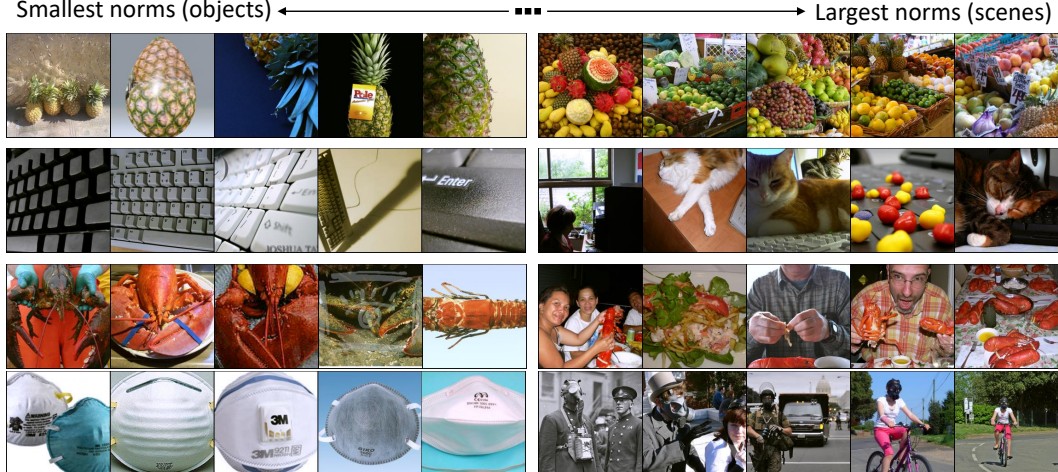

Figure 3: Images from ImageNet training set. The 5 images on the left have the smallest representation norms among all the images from the same class, and the 5 on the right have the largest norms.

have been recently identified [59, 54, 3], including labeling errors, multi-label images with only a single label provided, and so on. Although Beyer et al. [3] provide reassessed labels for the entire validation set, relabeling the entire training set can be infeasible.

Our learned representations provide a potential automatic way to identify images with multiple labels from datasets like ImageNet. Specifically, we first show in Figure 4 that there is a strong correlation between the representation norms and the number of labels per image according to the reassessed labels. For each class of the ImageNet training set, we rank the images according to their norms. The extreme images of some classes are shown in Figure 3 and also Appendix. Images with smaller norms tend to capture a single object, while those with larger norms are likely to depict a scene.

To quantitatively evaluate this property, we report the NDCG metric on the ranked images as shown in Table 3. NDCG assesses how often the scene images are ranked at the top. As a baseline, we rank the images based on the entropy of the class probability predicted by a classifier, which is a widely adopted label uncertainty indicator [11, 45]. We use both MoCo-v2 and supervised ResNet-50 as the classifier. As shown in Table 3, using norms with HCL achieves similar rank quality as using entropy with the supervised ResNet-50 on the ImageNet-ReaL dataset. In addition, when combining two ranks using simple ensemble methods such as Borda count, the score is further improved to $0.717$. This shows that the entropy and the norm might look at different aspects of the multi-label issue. For example, the entropy indicator can be affected by the bias of the model and the norm indicator can be wrong on the images with multiple objects from the same class. In addition, our method is dataset agnostic and does not need further training. To demonstrate this benefit, we report the same metric on the COCO validation, where we also have the number of labels for each image. Our method achieves much better NDCG scores than the supervised ResNet-50 as shown in Table 3. This finding can be potentially useful to guide label reassessment, or provide an extra signal for model training.

### 3.3.2 Out-of-Context Detection

Our hyperbolic loss $\mathcal{L}_{\text{hyp}}$ essentially encourages the model to capture the similarity between the object and scene. We further investigate this property on detecting the out-of-context objects, which can be useful in designing data augmentation for object detection [18]. We are especially interested in the out-of-context images with conflicting backgrounds. To this end, we use the out-of-context images proposed in the SUN09 dataset [15]. We first compute the representation of each object as well as the entire scene image with that object masked out. We then calculate the hyperbolic distance between the representations mapped to the Poincaré ball. Some example images from this dataset as well as the distance of each contained object are shown in Figure 5. We find that the out-of-context objects generally have a large distance, i.e. smaller similarity, to the overall scene image. To quantify this finding, we compute the mAP of the object ranking on each image and obtain $0.61$ for HCL. As a comparison, the MoCo similarity gives mAP $= 0.52$ and the random ranking gives mAP $= 0.44$.

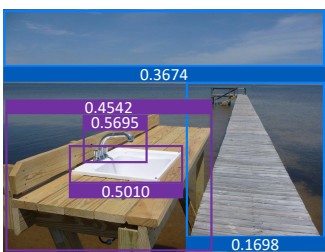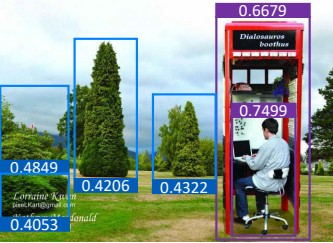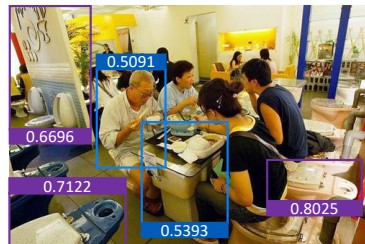

Figure 5: Out-of-context images from the SUN09 dataset [15]. The bounding box of each object, as well as its hyperbolic distance to the scene are displayed. The regular objects are in blue and the out-of-context objects are in purple. Note that the out-of-context objects tend to have large distances.

## 4 Main Ablation Studies

In this section, we report the results of several important ablation studies with respect to HCL. All the models are trained on the subset of the OpenImages dataset and linearly evaluated on the ImageNet-100 and our INPMix datasets. The top-1 accuracy is reported.

| Dist. | Center | IN-100 | IPS |
|-------|--------|--------|-----|
| - | - | 77.36 | 52.96 |
| Hyp. | Scene | 79.08 | 53.82 |
| Hyp. | Object | 76.96 | 52.74 |
| Euc. | Scene | 76.68 | 52.58 |

Table 4: Ablation on the similarity measure and hierarchy center.

| $\lambda$ | IN-100 | IPS |
|-----------|--------|-----|
| 0.01 | 77.70 | 53.43 |
| 0.1 | 79.08 | 53.82 |
| 0.2 | 78.64 | 53.84 |
| 0.5 | 0 | 0 |

Table 5: Ablation on the losses trade-off.

| Optim. | $\lambda$ | IN-100 | IPS |
|--------|-----------|--------|-----|
| RSGD | 0.1 | 79.08 | 53.82 |
| RSGD | 0.5 | 0 | 0 |
| SGD | 0.1 | 70.16 | 48.47 |
| SGD | 0.5 | 74.18 | 42.75 |

Table 6: Ablation on the RSGD versus SGD optimizers.

**Similarity measure and the center of the scene-object hierarchy.** We propose to use the negative hyperbolic distance as the similarity measure of the scene-object pairs. As an alternative, one can use cosine similarity on the hypersphere as the measure just like the original contrastive objective. However, this is basically minimizing the similarity between a single object and multiple objects. These objects are probably from different classes and hence conflict with the original objective. As shown in Table 4, replacing the negative hyperbolic distance with the Euclidean similarity impairs downstream performance. The resulting accuracy is even worse than the model without any loss function on the scene-object pairs. In terms of the hierarchy, we also test the assumption of scene-centric hierarchy [46, 47] by sampling the negative pairs as the objects and unpaired scenes. However, we notice a significant decrease in the downstream accuracy with this modification in Table 4.

**Trade-off between the Euclidean and hyperbolic losses.** We adopt the Euclidean loss to learn object-object similarity and the hyperbolic loss to learn object-scene similarity. A hyperparameter $\lambda$ is used to control the trade-off between them. As shown in Table 4, we find that a smaller $\lambda = 0.01$ leads to marginal improvement. However, we also observe that larger $\lambda$s can lead to unstable and even stalled training. With careful inspection, we find that in the early stage of the training, the gradient provided by the hyperbolic loss can be inaccurate but strong, which pushes the representations to be close to the boundary. As a result, the Riemannian SGD causes the gradient to be small and the training is consequently stuck at some the early point.

**Optimizer.** With the observation above, we ask whether RSGD is still necessary for practical usage. We replace the RSGD optimizer with SGD. To avoid the numerical issue when the representations are too close to the boundary, we increase $\varepsilon$ from $1e^{-5}$ to $1e^{-1}$. We first notice that this allows larger $\lambda$ to be used as opposed to the RSGD. However, SGD always yields inferior performance to RSGD. Therefore, it shows that the accurate gradient provided by RSGD is still necessary.

## 5 Related Work

**Representation Learning with Hyperbolic Space.** Representations are typically learned in Euclidean space. Hyperbolic space has been adopted for its expressiveness in modeling tree-like

structures existing in various domains such as language [58, 21, 51, 43, 44], graphs [2, 8, 9, 48], and vision [30, 10, 56]. The corresponding deep neural network modules have been designed to boost the progress of such applications [9, 22, 35, 55]. The hierarchical structure presented in the datasets can come from multiple factors, motivating the use of hyperbolic space. 1) Generality: the hypernym-hyponym property is a natural feature of words (e.g. WordNet [40]) and the hyperbolic space is extensively exploited to learn word embeddings that preserve that property [58, 21, 51, 43, 44]. Some image datasets also adopt the classes from WordNet for labeling, e.g. ImageNet [16], and consequently inherits the hierarchy in its labeling system. [36, 69, 38] take advantage of hyperbolic space to capture such information in the visual embeddings. 2) Uncertainty: Several studies have found that applying hyperbolic neural network modules to different tasks leads to a natural modeling of the uncertainty [23, 30, 56]. 3) Compositionality: The compositionality of different basic elements can form a natural hierarchy. We focus on learning the representations that capture the hierarchy between the objects and scenes. The hierarchical representations learned in the hyperbolic space have been applied to various tasks with the aforementioned motivations such as image classification [30] or segmentation [64, 23], zero-/few-shot learning [38, 36], action recognition [38], and video prediction [56]. In this paper, we aim at learning image representations for general purposes that can transfer to various downstream tasks.

**Self-Supervised Learning on Scenes.** Self-Supervised Learning (SSL) has made great strides in closing the performance with supervised methods [12, 14] when pretrained on the object-centric datasets like ImageNet. However, recent works have shown that SSL are limited on the multi-object datasets like COCO [52, 63] and OpenImages [41]. Several works have tried to address this issue by proposing different techniques. Dense-CL [63] works on pre-average pool features and uses dense features on pixel level to show improved performance on dense tasks such as semantic segmentation. DetCon [28] uses unsupervised semantic segmentation masks to generate features for the corresponding objects in the two views. CAST [53] uses GradCAM [52] to figure out same objects across views and applies contrastive loss on these features. PixContrast [68] uses pixel-to-propagation consistency pretext task to build features for both dense downstream tasks and discriminative downstream tasks. Pixel-to-Pixel Contrast [62] uses pixel-level contrastive learning to build better features for semantic segmentation. Self-EMD [37] uses earth mover distance with BYOL [25] for pretraining on the COCO dataset. ORL [67] uses selective search to generate object proposals, then applies object-level contrastive loss to enforce object-level consistency. ContraCAM [42] removes the scene bias issue by doing self-supervised object localization and performing contrastive loss on them. One of the reasons below-par performance of SSL methods can be attributed to treating scenes and objects using similar techniques, which often results in similar representations. In our work, instead of treating them in the same functionality, we use a hyperbolic loss, which builds representation that disambiguates scenes and objects based on the norm of the embeddings. Our method not only separates scenes and objects, but also helps us in improving downstream tasks such as image classification.

# 6   Closing Remarks

**Conclusion** We present HCL, a contrastive learning framework that learns visual representation for both objects and scenes in the same representation space. The major novelty of our method is a hyperbolic contrastive objective built on an object-centric scene hierarchy. We show the effectiveness of HCL on several benchmarks including image classification, object detection, and semantic segmentation. We also demonstrate the useful properties of the representations under several zero-shot settings from detecting out-of-context objects to quantifying the label uncertainty in the datasets like ImageNet. More generally, we hope this paper can encourage studies towards building a more holistic visual representation space and draw attention to the non-Euclidean representation learning.

**Limitations** Our model is shown to improve the classification performance on the ImageNet dataset, but not much on the more fine-grained classification tasks as shown in Appendix B.2. We conjecture that the largest improvement brought by our model to the object representations are modeling the context information, while most of these datasets share a general class whose contexts are more or less similar. In addition, although we provide some insights about the Riemannian optimization, its underlying mechanism in the visual representation learning is still not fully understood. We conduct more experiments on training hyperbolic linear classifiers in Appendix C.1. However, more efforts are needed to fully unleash the potential of non-Euclidean representation learning.

# 7 Societal Impact

Our work is a technical contribution and much of societal impact depends upon the models used in our work. We hope that our work will be used for betterment of the society and doesn't have any negative impact.

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
