# OpenReview forum: "Hyperbolic Contrastive Learning for Visual Representations beyond Objects"
_NeurIPS.cc/2022/Conference — NeurIPS 2022 Submitted_

### Official Review · Reviewer_L2gv · 2022-07-08

**Rating:** 5
**Confidence:** 4
**Soundness:** 2 fair
**Presentation:** 3 good
**Contribution:** 2 fair

**Summary:**

The authors address the problem of contrastive representation learning of objects and scenes simultaneously. To this end, they employ the Poincaré embedding to model a hierarchy from simple (single object) towards more complex (multi object, scenes) images.
In the experiment it is shown that HCL improves over MoCo-V2 [1]. The method is trained on COCO (118,000 images) or OpenImages (212,000 images) for 200 epochs.

**Questions:**

A more extensive comparison to other SSL methods would be necessary, especially to other methods that focus on object-level contrastive learning and/or that use object detection as additional information such as HCL.

**Limitations:**

Limitations are briefly addressed.

**Strengths And Weaknesses:**

Strengths:
- The idea of using hyperbolic geometry in conjunction with contrastive learning is interesting and promising.
- Results indicate the proposed method works: Quantitative improvements as well as representation norms correlating with number of labels.

Weaknesses:
- Performance gains of using hyperbolic loss are small and improvements might be due to using additional information, i.e. detections, instead of only the images.
- Relevant baselines are missing from Table 1. For detection for example [3, 4] achieve similar performance without relying on detection labels for pre-training.
- The claimed performance improvement is only shown over a weak baseline (MoCo-V2 on a small dataset). Performance in general does not seem competitive with current SSL models. For example, MoCo-V2 [1] achieves an accuracy of 67.5% when trained on ImageNet-1K for 200 epochs (71.1% after 800 epochs), the more recent DINO [2] even 75.3%. HCL achieves only an accuracy of 58.5% despite using the additional information of where objects are in an image. (This is of course difficult to compare due to using a different pre-training dataset, but a proper comparison to other SSL models would be needed here).
- The used datasets are small for SSL. To show that this method is useful in practice, I'd like to see some scores on a more competitive benchmarks. E.g. by using a dataset of similar size to ImageNet.
- Compared to other contrastive methods, HCL requires supervision. While selective search is proposed as a way to avoid reliance on ground truth, it decreases performance.

Minor remarks:
- "...treated using the same lens": This contradicts the claims of the next sentences
- Fig. 4 should indicate standard deviations.
- The precise task on the VOC dataset is unclear since there are multiple. I assume multi-label classification on Pascal VOC 2011 is meant but this should be explicitly stated.
- L189: MoCO-V2 should be trained using the hyperparameters from [1] and not the ones that work best for HCL (i.e. learning rate etc). MoCo-V2 is trained using a cosine scheduler which was apparently not being used here.


The paper presents an interesting idea of applying contrastive learning on multi-image scenes using poincare embeddings and promising first results. However, the experiments fail to convince me.
There is a large gap between this method's and state-of-the-art performance in self-supervised representation learning, which raises concerns regarding the utility of the method.

[1] Chen et al. Improved Baselines with Momentum Contrastive Learning
[2] Caron et. al. Emerging Properties in Self-Supervised Vision Transformers
[3] Wang et al. Dense Contrastive Learning for Self-Supervised Visual Pre-Training
[4] Xiao et al. Region Similarity Representation Learning

---

> ### Author Response · Authors · 2022-08-02
> **Reply to the Minor Remarks**
>
> 1 - 3: thanks for suggesting, we will clarify and add the corresponding details in our next version.
>
> 4: the general hyperparameters such as learning rates are tuned using MoCo-v2 instead of HCL. We did use a cosine learning rate scheduler during the pretraining phase.

---

> ### Author Response · Authors · 2022-08-02
> **Response to the Main Concerns**
>
> Thank you for your detailed feedback. To mitigate the raised concerns, we either conduct the requested experiments or explain the reasons why the suggested experiment may not be a fair comparison below.
>
> **"Performance gains of using hyperbolic loss might be due to using additional information, i.e. detections"**
>
> We agree that comparing with the original MoCo-v2 is not enough to demonstrate the effectiveness of using hyperbolic loss, therefore, in our experiments (e.g. Table 1), we also have another baseline HCL/$\mathcal{L}\_{\text{hyp}}$ that takes the object regions as the input while not using hyperbolic loss (Line 205-206). (We notice that the name of this variant HCL/$\mathcal{L}\_{\text{hyp}}$ does not explicitly reflect the fact that it uses bounding boxes and could have caused confusion. We will use a more accurate name like MoCo+bbox in our next version.) As shown in Tables 1 and 2, this baseline outperforms MoCo on the downstream tasks by learning cleaner object representations. However, without any objective imposed on the scene images, the performance is still inferior to the proposed HCL.
> We also report another ablation in Table 4 where we replace the hyperbolic distance in the loss function with a Euclidean distance, which leads to not just ineffective but detrimental results. We believe that these results together demonstrate that the improvements are indeed because of our novel objective rather than the detections.
>
> **"Dense-CL achieves similar performance without relying on detection labels for pre-training but is missing."**
>
> We train Dense-CL with and without our hyperbolic objective on the COCO dataset by following settings in the Dense-CL paper and evaluate the pretrained model on object detection and instance segmentation. To confirm that the improvements are due to the hyperbolic loss, we also report the results of Dense-CL with ground truth object proposals. The results are shown in the table below. We can see that using hyperbolic loss consistently improves the performance of Dense-CL.
>
> | COCO detection |  $\text{AP}$ |  $\text{AP}_{50}$ | $\text{AP}_{75}$ | COCO segmentation | $\text{AP}$ | $\text{AP}_{50}$ | $\text{AP}_{75}$ |
> |----------------------------|------|------|------|----------------------------|------|------|------|
> | Dense-CL                   | 39.6 | 59.3 | 43.3 | Dense-CL            | 35.7 | 56.5 | 38.4 |
> | Dense-CL+GT-Bbox   | 41.3 | 61.5 | 44.7 | Dense-CL+GT-Bbox  | 37.5 | 59.5 | 40.4 |
> | Hyperbolic Dense-CL | 42.5 | 62.5 | 45.8 | Hyperbolic Dense-CL | 38.5 | 60.6 | 41.4 |
>
> **"I'd like to see some scores on a dataset of similar size to ImageNet."**
>
> We train Dense-CL on the full OpenImage dataset for 75 epochs and the results are shown in the table below. The trend stays the same.
>
> | COCO detection |  $\text{AP}$ |  $\text{AP}_{50}$ | $\text{AP}_{75}$ | COCO segmentation | $\text{AP}$ | $\text{AP}_{50}$ | $\text{AP}_{75}$ |
> |----------------------------|------|------|------|----------------------------|------|------|------|
> | Dense-CL                   | 38.2 |  58.9 | 41.6 | Dense-CL            | 34.8 | 55.3 | 37.8 |
> | Dense-CL+GT-Bbox   | 41.1 | 61.5 | 44.4 | Dense-CL+GT-Bbox  | 37.2 | 58.3 | 39.7 |
> | Hyperbolic Dense-CL | 42.1 | 62.6 | 45.5 | Hyperbolic Dense-CL | 38.3 | 59.4 | 40.6 |
>
> **"HCL requires supervision. Selective search avoids annotations but it decreases performance."**
>
> We agree. However, unsupervised object discovery and detection is an active research area and has gained decent progress, e.g. [1]. We believe that using the proposals generated by these methods could further diminish the gap, which we leave as future work
>
> **"MoCo-V2 [1] achieves an accuracy of 67.5% for 200 epochs, 71.1% after 800 epochs, DINO [2] even 75.3%. HCL achieves only an accuracy of 58.5% despite using the additional information."**
>
> As also mentioned by the reviewer, these numbers are not directly comparable since these models have different focuses from us and are trained on the ImageNet-1K dataset which is object-centric.
>
> We have shown that the proposed HCL is generic and able to combine with any existing methods, including MoCo, ORL and Dense-CL as requested by reviewers. We also admit that our proposed idea has its limitations. For instance, we find that the learned representations do not improve on the fine-grained class classification as shown in Appendix B.2. However, we want to kindly note that as we state in the paper, our goal is ***not*** to develop another state-of-the-art self-supervised learning method but a step towards learning representations for images depicting not just objects. As we show in the paper, such representations allow us to achieve results that were not approachable before, such as using image-object similarity to detect out-of-context objects, and using representation norms to quantify label uncertainty.
>
> [1] Vo, Van Huy, et al. "Large-scale unsupervised object discovery." NeurIPS 2021.

---

> > ### Comment · Reviewer_L2gv · 2022-08-05
> > **Thanks for clarifying and additional experiments**
> >
> > Thank you for your clarifications and additional experiments.
> >
> > The authors added an additional comparison with DenseCL where they showed their method to improve COCO results, too, which strengthens the paper. Thus, I increase my rating.
> >
> >
> > My main concern remains that more recent SSL methods achieve much better classification accuracy on ImageNet and improvements of the hyperbolic loss compared to a MoCo+bbox baseline are small. The main increase in performance does seem to come from using object regions as the baseline using them increases significantly over MoCo compared to adding the additional loss. But I tend to agree that ImageNet, as an object-centric datataset, is not an ideal benchmark.
> >
> > If the main goal of the paper is to improve out-of-context detections, and quantifying label uncertainty, then the paper would benefit from a more detailed quantitative evaluation of them. For the claimed key features of their method, only limited quantitative evaluation is conducted (e.g. only comparison to MoCo).

---

### Official Review · Reviewer_78go · 2022-07-10

**Rating:** 6
**Confidence:** 4
**Soundness:** 3 good
**Presentation:** 2 fair
**Contribution:** 3 good

**Summary:**

This paper proposed a hyperbolic contrastive objective function (HCL) to formulate the object-centric scene hierarchy for self-supervised learning on scene images. The main motivation of the paper is that the hyperbolic space is more suitable to embed the hierarchical structure than
 Euclidean space. Using the ground truth or Selective search to obtain objects, this paper extends another branch on MoCo to constrain the similarity of scene crops and object crops in the hyperbolic space. The paper shows superior performance on downstream visual tasks when compared with its baseline method.  HCL also shows interesting properties on label uncertainty quantification and out-of-context object detection.

**Questions:**

1. Train HCL for 800 epochs and compare HCL with ORL and other Selective Search-based methods.  This result will be more important to support the claim of hyperbolic loss.
2. The paper claims in Line 213 that "HCL would improve more on the dataset with more objects per image", which is not sound enough. On OpenImages, the paper only compares HCL w/ and w/o hyperbolic loss when using the ground truth bounding boxes (which can not be acquired during self-supervised learning). The authors should provide the performance of HCL/${L_{hyp}}$ with SS to support the claim.
3. Is the additional hyperbolic loss time-consuming?

**Ethics Review Area:**

["I don’t know"]

**Limitations:**

Yes.

**Strengths And Weaknesses:**

Strengths:
1. The paper utilizes the hyperbolic space to formulate the scene-object relationship, which builds up another way to represent scene images (small norms represent objects and large norms represent scenes).
2. The paper achieves better downstream tasks than its baseline.

Weakness:
1. The experimental results only contain the comparison with baseline. Several relevant papers [1,2], which focus on the problem of self-supervised learning on scene images and also use Selective Search to acquire objects, are not mentioned in the experimental section.
2. The paper only trained HCL for 200 epochs, it is not clear whether the performance gain can be maintained when training for more epochs (800ep).


[1] Xie, Jiahao, et al. "Unsupervised object-level representation learning from scene images." Advances in Neural Information Processing Systems 34 (2021): 28864-28876.
[2] Li, Zhaowen, et al. "UniVIP: A Unified Framework for Self-Supervised Visual Pre-training." Proceedings of the IEEE/CVF Conference on Computer Vision and Pattern Recognition. 2022.

---

> ### Author Response · Authors · 2022-08-02
> **Response to Reviewer 78go**
>
> We would like to thank the reviewer for the positive comments and rating!
> 1. To address the major concern, we conduct an experiment using ORL. We train ORL with and without our hyperbolic objective on the COCO dataset following the settings in the paper (800 epochs) and evaluate it on object detection and instance segmentation. The results in the table below show that using our hyperbolic objective improves the performance. Similar to MoCo, methods like ORL only focus on learning object representations while paying insufficient attention to scene images. We show that it is beneficial to handle the representations for scene images additionally. We hope our work could make a step towards learning representations for all images beyond objects.
> | COCO detection |  $\text{AP}$ |  $\text{AP}_{50}$ | $\text{AP}_{75}$ | COCO segmentation | $\text{AP}$ | $\text{AP}_{50}$ | $\text{AP}_{75}$ |
> |-----------------|------|------|------|-----------------|------|------|------|
> | ORL         	| 40.3 | 60.2 | 44.4 | ORL         	|  36.3 | 57.3 | 38.9 |
> | Hyperbolic ORL | 41.4 | 61.4 | 45.5 | Hyperbolic ORL | 37.3 | 58.5 | 40.0 |
>
> 2. Please find our HCL/$\mathcal{L}_{\text{hyp}}$ with SS bbox on OpenImage below to support the claim.
> |                                | VOC   | IN-100 | INPMix | IN-1k |
> |--------------------------------|-------|--------|--------|-------|
> | HCL/$\mathcal{L}_{\text{hyp}}$ | 71.82 | 75.33  | 52.02  | 56.58 |
> | HCL                            | 74.31 | 78.14  | 53.21  | 58.12 |
>
> 3. Calculating hyperbolic loss itself takes nearly the same time as a normal contrastive loss. The only overhead in training is one additional forward pass to get scene representations.  We further measure the time per iterations during training: MoCo takes 0.616 sec/iter while HCL takes 0.757 sec/iter under 4 P6000 GPUs.

---

> > ### Comment · Reviewer_78go · 2022-08-09
> > **Thanks for the reponses**
> >
> > The author provides additional experimental results, showing the scalability of HCL on existing object-level learning methods (ORL), which supports their claims and address my main concerns. So, I increase my score.
> >
> > I would recommend the authors add these results to the main text to support the effectiveness of HCL.

---

> ### Author Response · Authors · 2022-08-07
> **A Gentle Reminder**
>
> Dear Reviewer 78go,
>
> Thank you for your time and efforts in reviewing our paper!
>
> We kindly remind that the discussion period will end in a few days, and thus we just wonder whether we could have the last chance to address your further concerns or questions (if you have any). We are sincerely glad to improve our paper under your suggestions!
>
> Thank you very much!
>
> Authors

---

### Official Review · Reviewer_SzXC · 2022-07-11

**Rating:** 6
**Confidence:** 3
**Soundness:** 3 good
**Presentation:** 4 excellent
**Contribution:** 3 good

**Summary:**

The paper aims to improve contrastive learning for multi-object scenes. In addition to making single-object representations close when objects are similar, the paper seeks to encourage multi-object scenes that share similar objects to also be close in the representation space. However, a given number of objects can be composed into exponentially many possible scenes, making it challenging to learn scene representations in the Euclidean space. Therefore, the proposed method maps the scene representations to a hyperbolic space, which is better at handling combinatorial explosion, and proposes a hyperbolic loss. Experiments show that the hyperbolic loss leads to 1~2% improvement in downstream classification, detection, and segmentation tasks. It is also shown that the learned scene representation can be used to quantify label uncertainty and detect out-of-context objects without further training.

**Questions:**

Please see my main concerns listed in Weaknesses. Additionally, I have the following two minor questions:
- I did not understand why the objects are the root nodes (rather than the scenes).
- Is there an intuitive explanation why the norm of representation will be large when there are multiple objects?

**Limitations:**

Limitations and potential negative societal impact have been addressed.

**Strengths And Weaknesses:**

- Strengths
    - SSL works well on images containing one dominant object, but has limitations on multi-object images. The paper tackles this limitation and is relevant to the community.
    - The proposed hyperbolic loss is quite novel and well motivated. It also leads to an interesting property of the representation that a larger norm indicates more object types in the image.
    - The paper is well written. It is easy for me to understand the high-level idea behind using the hyperbolic space although I am unfamiliar with the exact mathematical definitions.
- Weaknesses
    - Empirical evaluation seems insufficient. As mentioned in the Related Work section, quite a few models have been proposed to improve SSL on multi-object datasets, but none of them are included as a baseline. As an example, ORL[1] achieves 59% top-1 accuracy on ImageNet-1k when pretrained on COCO, while the proposed method only achieves ~55%.
    - Comparison to MoCo-v2 seems unfair, because MoCo-v2 only receives the full image as input while the proposed method takes object regions.
    - The definitions in L105-111 do not seem to form a tree. Consider an image $s$ with three objects $a$, $b$, and $c$, and a region $r$ covering objects $a$ and $b$. Then $a$, $r$, and $s$ form a cycle, so it is not a tree.

[1] Unsupervised Object-Level Representation Learning from Scene Images. (https://arxiv.org/pdf/2106.11952.pdf)

---

> ### Author Response · Authors · 2022-08-02
> **Response to the Main Concerns**
>
> **"Empirical evaluation seems insufficient..."**
>
> Although we build our method mainly on MoCo, the idea of hyperbolic objective can easily slot into other contrastive learning frameworks. For instance, as suggested by the reviewer, ORL [1] learns *object representations* from scene images and shows better downstream performance than MoCo when pretrained on COCO. We conduct experiments using ORL with and without our hyperbolic objective on the COCO dataset following the settings in [1], and evaluate the pretrained model on object detection and instance segmentation. We report the results in the table below. It shows that our proposed hyperbolic objective further improves ORL and supports our claim that learning representations for scene images in hyperbolic space is beneficial to downstream performance.
>
> | COCO detection |  $\text{AP}$ |  $\text{AP}_{50}$ | $\text{AP}_{75}$ | COCO segmentation | $\text{AP}$ | $\text{AP}_{50}$ | $\text{AP}_{75}$ |
> |---------|------|------|------|-----------------|------|------|------|
> | ORL         	| 40.3 | 60.2 | 44.4 | ORL         	|  36.3 | 57.3 | 38.9 |
> | Hyperbolic ORL | 41.4 | 61.4 | 45.5 | Hyperbolic ORL | 37.3 | 58.5 | 40.0 |
>
> Apart from ORL, as suggested by Reviewer-L2gv, we also add an experiment with Dense-CL [2]. Dense-CL is a contrastive learning framework that is not specifically designed for scene images but generally achieves better object detection results than MoCo. As shown in the table below, we observe improved performance when training Dense-CL with our hyperbolic objective. Together with the experiment on ORL, we believe that these results demonstrate that our proposed hyperbolic loss is generic enough to boost any existing contrastive learning methods.
>
> | COCO detection |  $\text{AP}$ |  $\text{AP}_{50}$ | $\text{AP}_{75}$ | COCO segmentation | $\text{AP}$ | $\text{AP}_{50}$ | $\text{AP}_{75}$ |
> |----------------------------|------|------|------|----------------------------|------|------|------|
> | Dense-CL                   | 39.6 | 59.3 | 43.3 | Dense-CL            | 35.7 | 56.5 | 38.4 |
> | Hyperbolic Dense-CL | 42.5 | 62.5 | 45.8 | Hyperbolic Dense-CL | 38.5 | 60.6 | 41.4 |
>
> **"Comparison to MoCo-v2 seems unfair..."**
>
> We agree that comparing with the original MoCo-v2 is unfair as it takes the full image as the input. We were aware of that, therefore, in our experiments (e.g. HCL/$\mathcal{L}\_\text{hyp}$ in Table 1), we gave MoCo advantages by using the object regions as the input (Line 205-206). We notice that the name of this variant HCL/$\mathcal{L}\_{\text{hyp}}$ does not reflect this explicitly and could have caused confusion. A more proper name should be MoCo+bbox. We will address this in our next version.
> This variant improves the downstream performance by learning cleaner object representations. However, without any objective on the scene images, the performance is still inferior to the proposed HCL. The other supporting evidence is in Table 4. When we replace the hyperbolic distance in the loss function with a Euclidean distance, we observe worse performance than MoCo. Therefore, the improvement is from leveraging the hyperbolic objective rather than having cropping.
>
> **"The definitions in L105-111 do not seem to form a tree."**
>
> Thanks for pointing this out! We agree that in this example it is not a tree so the reference to a forest is not accurate. We tried to refer to it as a hierarchy and will be more careful with this in our future version!
>
> [1] Xie, Jiahao, et al. "Unsupervised object-level representation learning from scene images." NeurIPS 2021.
>
> [2] Wang, Xinlong, et al. "Dense contrastive learning for self-supervised visual pre-training." CVPR. 2021.

---

> > ### Comment · Reviewer_SzXC · 2022-08-09
> > **Thanks for the additional experiments and explanation**
> >
> > In the additional experiments, the authors show that their proposed method is quite general and can be combined with many existing methods such as MoCo, ORL, and Dense-CL. The performance can be consistently improved by 1~3%.
> >
> > These results strengthen the paper, and have addressed my main concerns. The intuition provided by the authors also makes sense.
> >
> > While the performance improvement seems small, given the novelty and generality of the proposed method, I'd like to recommend accept.

---

> ### Author Response · Authors · 2022-08-02
> **Reply to the Minor Questions**
>
> **"why the objects are the root nodes"**
>
> To be accurate, when we mention objects we consider classes of objects, since in object-focused representation learning, different objects of the same class are often clustered together. In general, there are more scenes than classes of objects. A scene image can contain multiple kinds of objects and hence the number of scene images can be combinatorially many.  A real example of this is the COCO dataset, which has over 300K images but only less than 200 classes of objects. Last, objects are more general and ambiguous when shown alone in the image world. As more contexts are given, they become more specific. Each class of object can occur in different contexts and thus form a hierarchy starting from it. Therefore we pick the objects as the root.
>
> **"why the norm of representation will be large when there are multiple objects"**
>
> This is an expected result from our hyperbolic objective. Consider an example that learns representations of 3 regions $r_1$ (contain obj1, obj2, and obj3), $r_2$ (contain obj2, and obj3), and $r_3$ (contain obj2, obj3, and obj4): to minimize the hyperbolic distance $d(r_1, r_2) + d(r_2, r_3)$, it gives us the smaller distance when putting $r_2$ at the origin and $r_1$ & $r_3$ at the boundary compared with putting $r_1$ or $r_3$ at the origin. This is because of equation (3), which tells us that the distance between two boundary points is twice the distance between one to the origin. Therefore, the hyperbolic loss places regions with more objects closer to the periphery, with larger norms.
>
> We also experimented with the *scene*-centric hierarchy as shown in Table 4. Qualitatively, we saw that the model overfits to make the norms of some scene regions to be smaller but they are not consistent across images and do not generalize to unseen images. Quantitatively, it yields worse performance. We appreciate that the reviewer asks these questions on the intuition and will make our best effort to improve on this in future versions.

---

> ### Author Response · Authors · 2022-08-07
> **A Gentle Reminder**
>
> Dear Reviewer SzXC,
>
> Thank you for your time and efforts in reviewing our paper!
>
> We sincerely hope our response successfully addresses your concerns and answers your questions.
>
> We just wonder if we could have the last chance to address your further concerns or questions, and we are glad to improve our paper under your further suggestions (if you have any).
>
> Thank you very much!
>
> Authors

---

### Author Response · Authors · 2022-08-02
**General Response**

We thank the reviewers for their thoughtful feedback. It is encouraging that the reviewers found learning representations for scene images to be important and relevant to the community (R-SzXC), our idea of using hyperbolic space to be novel (R-SzXC), interesting and promising (R-L2gv), and well-motivated (R-78go, R-SzXC), which is corroborated by our improved downstream performance (R-78go, R-L2gv) and the application of the useful property  of representation norms to several problems, and our writing to be clear and easy to follow (R-SzXC).

---

### Meta-Review · Area_Chair_Q81N · 2022-08-30

**Recommendation:** Reject
**Confidence:** Certain

**Metareview:**

Overall, reviewers found that the method is sound but the results are marginal.

There are numerous frameworks for self-supervised learning today. The one introduced here underperforms compared to others, like ORL and Dense-CL, as pointed out by the reviewers. The authors in their response then combined their method with ORL and Dense-CL. This resulted in 1.1% improvement over ORL. This is marginal. I understand that the authors intended to reposition their work as a general-purpose add-on that increases performance. But, far more analysis is required to establish that this 1.1% improvement is real and that is meaningful. There are many tricks for SSL that improve performance by 1% or so, often these are not used because the slowdown they incur is not worth the effort. And the increase in performance is not noticeable.

At least, if the authors wish the pivot in this way, then the manuscript requires a rewrite to read as an add-on to many methods and to properly evaluate this. As it stands, by not comparing against ORL and other methods in the main manuscript, the submission cannot be accepted as is.

I encourage the authors to submit to a computer vision venue where such results may be appreciated more; where they may be given more room to thrive into something bigger. And to fully incorporate methods like ORL while demonstrating that their method really does produce a meaningful improvement.


**Award:**

No

---

### Decision · Program_Chairs · 2022-09-14

Reject